# Towards Agentic Self-Learning LLMs in Search Environment

## Abstract

We study whether self-learning can scale LLM-based agents without relying on human-curated datasets or predefined rule-based rewards. Through controlled experiments in a search-agent setting, we identify two key determinants of scalable agent training: the source of reward signals and the scale of agent task data. We find that rewards from a Generative Reward Model (GRM) outperform rigid rule-based signals for open-domain learning, and that co-evolving the GRM with the policy further boosts performance. Increasing the volume of agent task data—even when synthetically generated—substantially enhances agentic capabilities. Building on these insights, we propose **Agentic Self-Learning** (ASL), a fully closed-loop, multi-role reinforcement learning framework that unifies task generation, policy execution, and evaluation within a shared tool environment and LLM backbone. ASL coordinates a Prompt Generator, a Policy Model, and a Generative Reward Model to form a virtuous cycle of harder task setting, sharper verification, and stronger solving. Empirically, ASL delivers steady, round-over-round gains, surpasses strong RLVR baselines (e.g., Search-R1) that plateau or degrade, and continues improving under zero-labeled-data conditions, indicating superior sample efficiency and robustness. We further show that GRM verification capacity is the main bottleneck: if frozen, it induces reward hacking and stalls progress; continual GRM training on the evolving data distribution mitigates this, and a small late-stage injection of real verification data raises the performance ceiling. This work establishes reward source and data scale as critical levers for open-domain agent learning and demonstrates the efficacy of multi-role co-evolution for scalable, self-improving agents. The data and code of this paper are released at https://anonymous.4open.science/r/Towards-Agentic-Self-Learning-4D63

## 1 Introduction

Autonomous agents built on Large Language Models (LLMs) have evolved rapidly, from early prompt-engineering pipelines to recent reinforcement learning–trained systems capable of self-improvement in interactive environments. Early work such as ReAct-style reasoning–acting frameworks and reflective methods constructed agent loops through manually designed prompt templates, memory modules, and decision-making heuristics (?Yao et al., 2022; Sun et al., 2023). These policy-evolution approaches refine the agent's reasoning process via self-reflection, co-adaptation, or prompt rewriting, and can effectively integrate with other inference-time enhancing technique such as Retrieval-Augmented Generation (RAG).

A second wave of research extends the RL with Verifiers and Rewards (RLVR) paradigm, pioneered in reasoning-focused models such as OpenAI o1 (Jaech et al., 2024) and DeepSeek-R1 (Guo et al., 2025), to the agent setting. Works like WebAgent-R1 (Wei et al., 2025), Search-R1 (Jin et al., 2025) and R1-Searcher (Song et al., 2025) train LLM-based agents end-to-end with reinforcement learning in interactive environments through rule-based reward signals, enabling them to continuously refine planning, tool use, and long-horizon strategies without relying solely on static instruction-tuned datasets.

In parallel, there has been a growing trend toward *self-learning Large Reasoning Models (LRMs)*, which aim to improve reasoning ability through reinforcement learning, but without relying on large quantities of annotated human data. Within this paradigm, methods such as Absolute-Zero (Zhao

et al., 2025) and R-Zero (Huang et al., 2025) can achieve self-evolution LRMs under zero-labeled-data conditions.

However, most existing approaches still rely heavily on human-annotated datasets or rule-based environments (e.g., code interpreters) to provide verifiable rewards within the RLVR paradigm. Such dependence significantly constrains the potential of LLM agents to scale the self-learning paradigm to open-domain settings, where reliable, automatically verifiable feedback is scarce and task structures are less formalized.

Motivated by these developments, we seek to explore whether the self-learning paradigm can be effectively extended to scale the training of agentic systems, **freeing them from reliance on both human-curated datasets and predefined rule-based reward functions**. To this end, we begin by conducting controlled experiments in the *search-agent* setting introduced by SEARCH-R1 (Jin et al., 2025), in order to investigate two key factors for LLM-based agentic reinforcement learning: the *source of reward signals* (§3.2) and the *scale of agent task data* (§3.2). Our findings reveal that:

- Compared to rigid, rule-based rewards, signals produced by a *Generative Reward Model* (GRM) are more effective for scaling LLM agents to open-domain scenarios. Moreover, when the GRM is co-evolved with the policy model, thereby progressively acquiring stronger generative discrimination capabilities, agent performance improves further.

- Increasing the quantity of agent task data — even when tasks are synthetically generated — can substantially enhance LLMs' agentic capabilities.

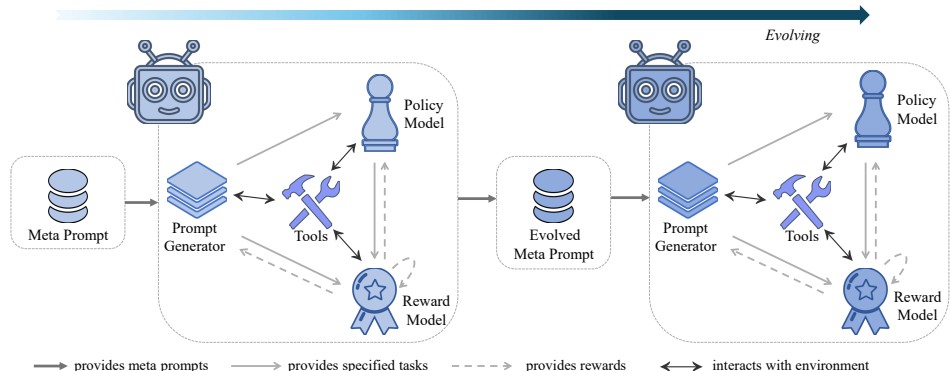

Figure 1: The framework of Agentic Self-Learning (ASL).

Based on these observations, we present Agentic Self-Learning (ASL) — a fully closed-loop, multi-role reinforcement learning framework that unifies task generation, policy execution, and evaluation within a shared tool environment and LLM backbone. ASL is built upon three mutually reinforcing functional roles: the *Prompt Generator*, the *Policy Model*, and the *Generative Reward Model*. Through iterative optimization of these three roles, ASL embodies an agentic, self-improving loop: generating diverse and adaptively challenging tasks, solving them, and refining evaluation criteria in tandem. This design not only mitigates the need for external supervision but also enables scalable, open-domain learning that is robust against reward hacking.

To empirically evaluate the efficacy of ASL, we conduct a series of experiments to answer the following three questions:

**Can ASL effectively train the LLM agent compared to existing methods?** ASL delivers steady, iteration-over-iteration gains and ultimately surpasses strong RLVR method such as Search-r1, which excels early but quickly plateaus and degrades. Other self-learning methods (e.g., Absolute Zero, R-zero) also show early gains followed by stagnation. In contrast, ASL continues improving even under zero-data conditions, highlighting its superior sample efficiency and robustness.

**Does the three roles in ASL co-evolve during the training process?** We observe a clear closed-loop synergy: the Prompt Generator produces progressively harder questions, the Generative Reward Model improves its verification accuracy each round, and the Policy Model steadily increases

task accuracy. This virtuous cycle—harder question-setting, sharper verification, and stronger solving—confirms coordinated co-evolution.

**What factors might limit the ongoing evolution of ASL and how can we mitigate them?** The main bottleneck is the GRM's verification capacity. When the GRM is not updated, the Prompt Generator learns to exploit scoring blind spots, triggering reward hacking and stalling Policy Model progress. Continual GRM training on the evolving distribution mitigates this failure mode and sustains improvement over longer horizons. Moreover, injecting a small amount of real verification data further lifts the performance ceiling, indicating that ASL's upper bound is effectively set by GRM capability. Practically, a two-phase strategy works well: rely on self-generated data to continually calibrate the GRM, then apply a modest real-data calibration late in training to refresh the ceiling and unlock additional gains.

In summary, this paper makes the following foundational contributions:

- Through controlled experiments, we identify the *source of reward signals* and the *scale of agent task data* as two critical factors for scaling LLM agents in open-domain settings.
- We propose ASL, the first multi-role closed-loop agentic self-learning framework, enabling simultaneous co-evolution of task generation, problem solving, and evaluation.
- Through experiments, we demonstrated that ASL can effectively coordinate the improvement of problem generation, solving, and verification capabilities over multiple rounds of iteration; meanwhile, our validation results revealed that the capability boundary of GRM is an important factor limiting the upper bound of ASL, and we further proved that this upper bound can be raised by training GRM with either synthetic or real data.

## 2 RELATED WORK

**Policy Evolution.** Currently, the development of self-evolving agents focuses primarily on two key aspects. One area emphasizes the innovation of internal learning mechanisms to drive agent evolution. This aspect is centered on optimizing the agent's internal learning, decision-making, and adaptation processes, typically without relying on structured adjustments to the external task environment. For instance, Reflexion (Shinn et al., 2023) introduces a natural language feedback mechanism that replaces traditional weight updates, enhancing decision-making capabilities through error-driven self-reflection memory. Mutual-Taught (Shi et al., 2025b) employs the Expectation-Maximization (EM) algorithm to jointly optimize the policy model and reward model, allowing for collaborative adaptation to distribution shifts. AgentEvolver (Belle et al., 2025) adopts a multi-role collaboration mechanism, which enables agents to rewrite prompts and decision codes independently without human intervention and further promotes the autonomous optimization of internal mechanisms. The essence of these methods lies in improving the agent's abilities through enhancements in internal training algorithms, feedback mechanisms, or data generation strategies.

**Task Adaptation.** Another research focus is the dynamic evolution of task difficulty and structure. A meticulously designed task environment serves as a powerful driving force for evolution (Gao et al., 2025). AlphaEvolve (Novikov et al., 2025) proposes an evolutionary code generation framework that utilizes multi-round collaboration among large language models (LLMs) and difficulty cascading to optimize algorithm design while the Self-Challenging Agent (SCA, Zhou et al. 2025) introduces a "code-as-task" framework that achieves a leap in tool invocation capabilities through the generation of self-verifiable tasks. TaskCraft (Shi et al., 2025a) is dedicated to the automatic generation of complex multi-tool combination tasks, systematically honing the reasoning and planning abilities of the agents. While these methods have demonstrated significant effectiveness in structured tasks and coding, they often fall short in addressing complex semantic understanding and judgment scenarios in open-domain question-answering texts. To bridge this gap, our work presents a self-evolving agent oriented toward question-answering texts, with the core innovation of using distribution entropy as both a measure of difficulty and an evolution reward signal, thereby constructing a closed-loop task evolution engine.

**Agent-R1.** Traditional large language models (LLMs) rely on human-designed workflows. In contrast, end-to-end reinforcement learning (RL) methods empower agents to autonomously make decisions and perform actions by learning optimal policies through interactions with dynamic online environments. WebAgent-R1 (Wei et al., 2025) implements multi-round interactive end-to-end

optimization to support scalable online interactive learning within complex network environments. NB-Agent (Zhang et al., 2025) integrates a code-driven interaction paradigm with distributed reinforcement learning training, and further introduces sandbox-based acceleration techniques and memory optimization strategies to achieve efficient large-scale online training. Furthermore, Absolute Zero (Zhao et al., 2025) and R-Zero (Huang et al., 2025), which leverage Reinforcement Learning with Verifiable Rewards (RLVR), generate and solve tasks respectively through self-play mechanisms and two-role adversarial evolution, breaking through the dependence on human-labeled data and promoting the self-evolution. The deep integration of reinforcement learning with intelligent agents opens up new possibilities for the autonomous completion of long-term and complex tasks.

# 3 AGENTIC SELF-LEARNING

We propose **Agentic Self-Learning** (ASL), a multi-agent self-improvement framework designed to enable large language models (LLMs) to continuously evolve their reasoning, generation, and evaluation capabilities in an autonomous closed loop. We first introduce two key factors we have identified that influence RL training for LLM agents: the scale of agent task data and the source of reward signals (§3.2). We then present the three key components of the ASL framework—the prompt generator, the policy model, and the reward model (§3.3), and finally describe the training procedure of the ASL framework (§3.4).

## 3.1 PRELIMINARIES AND EXPERIMENT SETTINGS

**Task and Data.** We focus on applying ASL to the Deep Search task. Given a knowledge-intensive question $x$, the agent is supposed to interact with the search engine for multiple turns, and finally derive its answer $\hat{y}$, which will be match with the ground-truth answer $y$ to determine correctness. We conduct evaluation on seven widely-used benchmark datasets, covering both general and multi-hop question answering scenarios. The **general QA** category includes *Natural Questions* (NQ) Kwiatkowski et al. (2019), *TriviaQA* (Joshi et al., 2017), and *PopQA* (Mallen et al., 2022). The **multi-hop QA** category consists of *HotpotQA* (Yang et al., 2018), *2WikiMultiHopQA* (Ho et al., 2020), *MuSiQue* (Trivedi et al., 2022), and *Bamboogle* (Press et al., 2022). These datasets span a diverse set of search and reasoning challenges, enabling a comprehensive assessment of model performance across both in-domain and out-of-domain settings.

**Models and Framework.** Following the experimental setup adopted in SEARCH-R1 (Jin et al., 2025), we conduct experiments with Qwen-2.5-7B-Instruct. For the retrieval component, the 2018 Wikipedia dump (Karpukhin et al., 2020) is used as the knowledge corpus, and we employ the E5 retriever (Wang et al., 2022). The system prompts that used for format and tool description can be found in Appendix A. The number of retrieved passages is fixed to 3 across all methods. For the training framework, we implement the agentic reinforcement learning component based on VeRL (Sheng et al., 2024). The user prompts and generation examples of three roles can be found in Appendix A,B.

## 3.2 EMPIRICAL MOTIVATION FOR ASL DESIGN

Before finalizing the ASL architecture, we conducted two preliminary studies to investigate key design choices in agentic RL with verifiable rewards, which directly motivate our proposed multi-role closed-loop framework. Following the settings in Search-R1 (later introduced in §3.1), we select Qwen-2.5-7B-instruct as the base model and train and test the policy models in search-based question-answering environment. We first conducted the following two controlled experiments to identify the *source of reward signals* (§3.2) and the *scale of agent task data* (§3.2) as two critical factors for scaling LLM agents in open-domain settings.

**(1) The Source of Reward Signals.** In RLAIF-style settings, the choice of reward function critically influences policy optimization. We compared the following three variants:

- **Rule-based Reward**: answers are judged using substring exact match with gold answers.
- **Fixed Generative Reward Model**: the base policy model (reference model) acts as a agentic judge producing rewards.

- **Policy as Reward Model**: the generative reward model shares parameters with the policy model so the generative reward model can generalize from improvements in the policy model's agent capabilities to achieve stronger agentic verification ability.

As shown in Figure 2, both generative reward settings outperform purely rule-based rewards, confirming that a generative judge can provide more informative and tolerant reward signals. Furthermore, the GRM sharing parameters with the policy model achieves the highest and keeping growing accuracy, suggesting that generative reward models with stronger agentic verification ability are even better reward functions for LLM agentic RL. This empirical evidence underpins our decision to include a trainable policy-shared GRM as a core role in ASL.

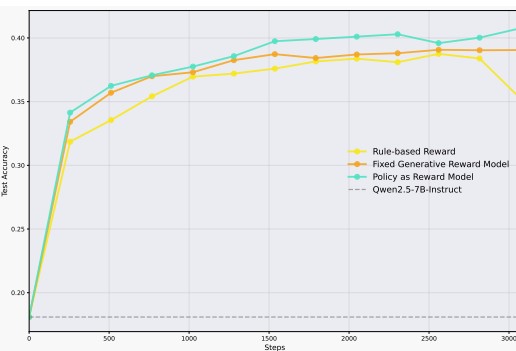

Figure 2: Performance comparison between rule-based, fixed GRM, and policy-shared GRM reward functions. A co-evolving GRM yields the best reward quality and downstream policy performance.

**(2) The Scaling of Agentic Data.** We next investigated whether scaling the amount of *agent-generated* training data benefits policy learning when using a GRM as the reward function. Starting from the same meta prompts (Appendix A), we generate varying quantities of search-based QA tasks (1k, 10k and 46k examples) and accordingly train the LLM agent on them.

Results in Figure 3 show a consistent improvement with increased generated data volume. This finding motivates ASL's closed-loop Prompt Generator design — by autonomously producing high-quality, diverse tasks at scale, ASL continuously supplies the policy with valuable new training data without human annotation cost.

These two findings jointly justify the ASL multi-role design: (1) a co-evolving GRM is superior to fixed or rule-based rewards, and (2) enabling the agent to scale high-quality, verifiably judged tasks is a viable path to sustained performance growth.

## 3.3 CONSTITUENT ROLES

**Prompt Generator (PG).** Conditioned on a *meta prompt*, the Prompt Generator is responsible for producing new training tasks in the form of problem–solution pairs $(x, a)$, where $x$ denotes the synthesized question and $a$ denotes its reference or ground-truth answer. In addition to producing initial tasks, the Prompt Generator evolves the task distribution over iterations, thereby adaptively increasing task complexity to match the model's learned capabilities.

**Policy Model (PM).** Given a task $x$, the Policy Model generates a candidate solution $y$. This role essentially models the problem-solving capability of the system and is the primary target for performance improvement. Its outputs are subsequently evaluated by the Generative Reward Model to obtain scalar feedback signals.

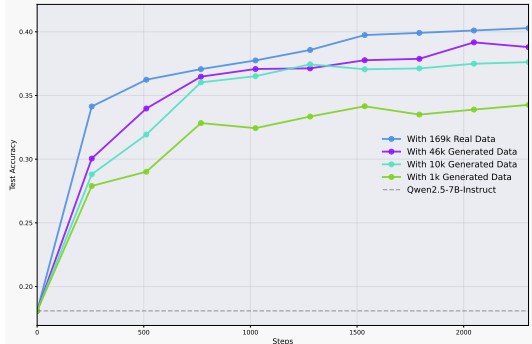

Figure 3: Effect of scaling the amount of GRM-rewarded agentic data on downstream accuracy. Larger amounts of verifiably judged agent-generated data substantially improve policy performance.

**Generative Reward Model (GRM).** Provided with a problem $x$ and a generated solution $y$, the Generative Reward Model produces a correctness score $s \in \{0, 1\}$. The generative reward model itself is also trained via RL to improve its ability to faithfully and consistently assess outputs.

## 3.4 TRAINING PROCESS

ASL operates in an iterative cycle that progresses through RL training the three roles in sequence: PG $\rightarrow$ GRM $\rightarrow$ PM $\rightarrow$ PG .... At each stage, one role is actively optimized while the others provide fixed contextual behavior within the loop. Assuming we are at the $t$th iteration:

**Phase 1: Prompt Generator Training.** Given a meta prompt $u$ from PG training data $D_{PG}^{(t-1)}$ that is constructed in $t - 1$ iteration, the PG synthesizes a batch of $N$ candidate problem–answer pairs $\{(x_n, a_n)\}_{i=1}^N$. Then, for illustrative purposes, consider a single pair $(x, a)$ in these generated questions.

As the PG may generate questions that beyond the verification ability of the GRM, we first evaluate its validity using the current GRM: the GRM performs $K$ independent rollouts to assess the correctness of $(x, a)$, producing a set of scores $\{s^k\}_{k=1}^K$. The average score $\bar{s}$ is compared against a predefined threshold $C$. Only if $\bar{s} > C$ is $(x, a)$ retained as a valid problem instance, otherwise, $(x, a)$ will be discarded.

For each valid $(x, a)$, we then store the pair into the PM training set $D_{PM}^{(t)}$, ensuring that high-quality tasks contribute to downstream policy learning. Additionally, to compute the reinforcement reward signal for the PG, we let the current PM attempt to solve $x$ via $M$ rollouts, yielding a set of responses $\{y_m\}_{m=1}^M$. The GRM then scores each response, resulting in corresponding verification scores $\{s_m\}_{m=1}^M$. These scores play a dual role:

- We add the triples $\{(x_m, y_m, s_m)\}_{m=1}^M$ to the GRM's training data $D_{GRM}^{(t)}$ of $t$th iteration.
- We measure the entropy of this score distribution (i.e., the multiple attempts at answering question $x$), $r_{\mathrm{PG}} = \mathrm{Entropy}(s_1, s_2, \ldots, s_m)$ and use it as the reward signal to update the PG's parameters. Intuitively, higher entropy indicates that the task elicits diverse performance from the PM, and will guide the PG to produce more informative and discriminative task design.

**Phase 2: Generative Reward Model Training.** In this phase, the GRM is optimized to produce evaluation scores that are both consistent and faithful to verifiable correctness. Given a problem–solution pair $(x, y)$ the GRM performs $N$ independent rollouts, yielding a set of predicted scores $\{\hat{s}_n\}_{n=1}^N$. For each predicted score $\hat{s}$, we compare it against the reference score $s$ generated in Phase 1. This comparison produces a binary correctness indicator $\mathbf{1}(\hat{s} = s)$ for each rollout, reflecting whether the GRM's judgment matches the reference evaluation.

This binary correctness signal is then served as the reward for GRM $r_{GRM}$ and used to update the GRM's parameters via reinforcement learning, following the RLVR (Reinforcement Learning with Verifiable Rewards) paradigm. By grounding its learning signal in rule-based verification, the GRM incrementally improves its ability to assign precise and reliable scores to diverse model outputs, which in turn strengthens downstream policy optimization.

**Phase 3: Policy Model Training.** In this phase, the PM is trained to improve its problem-solving ability on tasks generated by the PG. For a given problem $x$, the PM performs $N$ independent rollouts, producing a set of candidate answers $\{y_n\}_{n=1}^N$. Each answer is then evaluated by the GRM, yielding a corresponding set of scores $\{s_n\}_{n=1}^N$. These scores play a dual role:

- Reinforcement signal for the PM — the collection $\{s_n\}_{n=1}^N$ is directly used as the reward signal $r_{PM}$ in the PM's RL optimization objective, encouraging it to maximize expected performance over the task distribution.
- Difficulty feedback for the PG — we compute the average score $\bar{s} = \frac{1}{N} \sum_{n=1}^N s_n$. If $\bar{s} > 0.5$, the task is considered too easy for the current PM; we then assign a difficulty flag $f = \mathrm{HARDER}$, indicating that the PG should increase task difficulty in subsequent iterations. Conversely, if $\bar{s} \leq 0.5$, the task is considered too challenging, and we set $f = \mathrm{EASIER}$, signaling that the PG should generate simpler variants. Finally, each triplet $(x, a, f)$, where $a$ is the reference answer of $x$, is stored into the PG's training dataset $D_{PG}^{(t)}$ for use in the next iteration.

## 3.5 CLOSED-LOOP EVOLUTION

ASL will continuously cycle through the three training phases above for self-learning. Phase 1 will stop when the number of $D_{PM}^{(t)}$ reaches a certain threshold, and Phases 2 and 3 will stop

when their respective training rewards converge. All three roles in ASL share the entire LLM parameters. The other detailed setting can be viewed in Appendix D Through the cyclic optimization of these three roles, ASL implements a self-reinforcing curriculum: the PG gradually proposes more challenging tasks, the PM adapts to solve them more effectively, and the GRM continuously refines its evaluation criteria. Since all roles share parameters, skill improvements in one role naturally benefit the others, accelerating overall capability enhancement. Over time, ASL enables the LLM to autonomously expand both its problem-solving range and evaluation robustness without any direct human supervision.

## 4 EXPERIMENTS

In this section, we first introduce the settings used in our experiments (§3.1), then we conducted a series of experiments to answer the following three research questions:

- **RQ1:** Can ASL effectively train the LLM agent compared to existing methods? (§4.1)

- **RQ2:** Does the three roles in ASL co-evolve during the training process? (§4.2)

- **RQ3:** What factors might limit the ongoing evolution of ASL and how can we mitigate them? (§4.3)

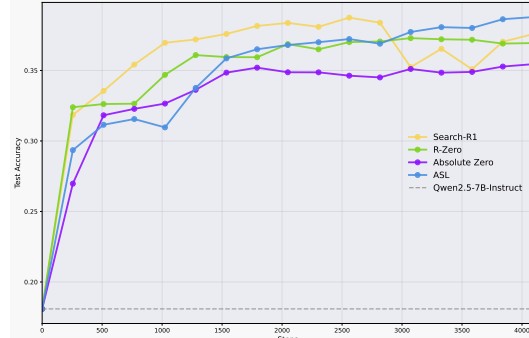

Figure 4: The test accuracy of LLM agents trained by ASL and baseline methods.

### 4.1 ASL VS BASELINES

We first compare ASL with other agent training methods:

- **Search-R1** (Jin et al., 2025) Search-R1 directly brought the RLVR paradigm used in OpenAI o1 (Jaech et al., 2024) and DeepSeek-R1 (Guo et al., 2025) to the training scenario of search agent.

- **Absolute Zero** (Zhao et al., 2025) Absolute Zero proposed a Proposer-Solver self-play framework. It proposes to encourage Proposer to generate low (but non-zero) success rate questions and verify the correctness of Solver through Python execution. We implement the Absolute Zero training in our agentic search-based QA setting.

- **R-Zero** (Huang et al., 2025) R-Zero proposed a Challenger-Solver co-evolving framework. It leverages an uncertainty reward and repetition penalty to guide the training of Challenger and trains the Solver through RLVR. We implement the R-Zero training in our agentic search-based QA setting.

As shown in Figure 4, Search-r1, an RLVR approach trained on real data, achieves the best performance early in training but then quickly converges and degrades due to poorer generalization. Absolute Zero and R-zero show similar behavior: they make substantial gains in the first two iterations with Proposer/Challenger generating harder questions are Solver effectively learn from the tasks, but they then stall and enter a plateau in the third iteration and later. In contrast, ASL steadily improves the agent's capability over multiple iterations and ultimately surpasses search-r1 even under zero-data conditions.

### 4.2 SYNERGISTIC CO-EVOLUTION OF THE THREE ROLES

We track how the three roles in ASL—PG, GRM, and PM evolve across iterations. As shown in Figure 5, the closed-loop training leads to the following coordinated improvements:

**Question generation strength**: In each iteration, we sample 2,000 generated questions and have the same fixed evaluator (Search-r1 trained for 1,024 steps) solve them. The blue bars plot this accuracy. As iterations progress, the accuracy generally declines, indicating that the generator produces increasingly challenging problems that better stress a fixed solver.

**Verification strength**: We build a verification test set from the original test set by representing each ground-truth pair $(q, a^+)$ as $(q, a^+; correct)$, and using GPT-4o-mini to synthesize a wrong answer $a^-$, yielding $(q, a^-; wrong)$. The green bars show the GRM's accuracy on this verify set, which rises with each iteration, evidencing stronger discrimination between correct and incorrect solutions.

**Solving strength**: The yellow bars report the PM's task solving accuracy. This steadily increases across iterations, showing that the solver benefits from harder questions and sharper supervision from the verifier.

Together, these trends demonstrate that ASL drives a virtuous cycle: stronger question-setting and stricter verification improve the training signal, which in turn elevates solving performance; the improved solver and verifier then enable the generator to craft even harder questions. ASL thus advances question generation, judging, and solving in tandem.

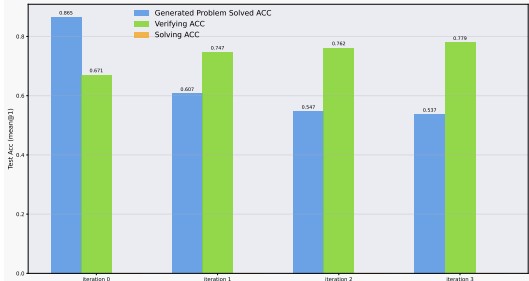

Figure 5: Comparison across iterations. Blue: accuracy of a fixed evaluator (Search-r1, trained 1,024 steps) on 2,000 questions generated in each iteration—lower values indicate harder generated problems. Green: GRM accuracy on the verify set constructed from $(q, a^+; correct)$ and GPT-4o-mini–derived negatives $(q, a^-; wrong)$—higher is better judgment. Yellow: PM solving accuracy—steadily improving. These patterns show the three roles co-evolve under ASL.

### 4.3 IDENTIFYING AND LIFTING THE CEILING OF ASL

**Identify reward hacking.** An important factor limiting RL scaling is reward hacking. To investigate whether the hacking of PM towards GRM is the bottleneck in ASL's evolution, we ablate the GRM training phase and plot the resulting training and testing curves (Figure 6a). In the early iterations, the system exhibits a healthy co-evolutionary dynamic: the PG progressively proposes harder problems, while the PM correspondingly improves its solving ability during its training phase. Each time the training stage switches from one component to the other, we observe a sharp drop in rewards. This drop reflects the sudden increase in the counterpart's capability—PG confronts a stronger solver, and PM faces more challenging prompts—exactly the kind of "difficulty escalates, ability catches up" rhythm one expects from a well-structured training loop.

However, by the third cycle a failure mode emerges. Because the GRM is not explictly trained and therefore has limited verification ability, the PG learns to generate problems that are excessively difficult or even unsolvable. These prompts lie far outside the GRM's calibration range, causing its scores on such out-of-distribution inputs to become effectively random. The resulting high uncertainty inflates the entropy-based reward signal, which the PG then exploits to obtain spuriously high returns. This is a textbook instance of reward hacking: the reward becomes decoupled from true problem quality or solvability, and the PG is incentivized to game the scoring mechanism rather than improve the curriculum.

Consequently, the PM cannot extract meaningful learning signal from unsolvable prompts and fails to make further progress, leading to a plateau in test performance. Taken together, these dynamics indicate that a non-updating GRM not only becomes the bottleneck for ASL's evolution but also opens the door to exploitative behaviors, underscoring the necessity of co-training or continual calibration of the reward model to track the evolving data distribution.

**Mitigating reward hacking.** When we reintroduce the GRM training phase into the ASL loop, the early reward hacking behavior is substantially reduced (Figure 6b). Co-training the GRM with the evolving distribution of prompts and solutions stabilizes its scoring on harder inputs, dampens spurious entropy rewards, and allows ASL to make steady progress for a longer training horizon. In other words, as PG proposes more challenging problems and PM adapts, a concurrently updated GRM continues to calibrate its judgments, preventing the PG from exploiting scoring blind spots and preserving a meaningful learning signal for PM.

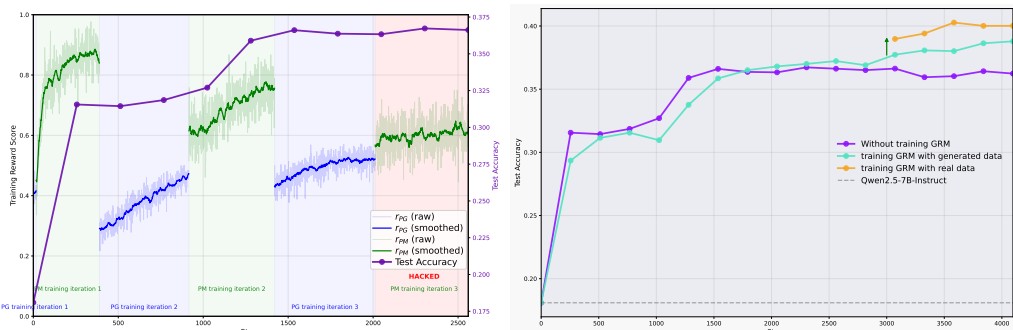

(a) The training and testing curve of ASL while GRM is NOT trained. The system exhibits a healthy co-evolutionary dynamic in early iterations but then trapped in reward hacking.

(b) The test accuracy of LLM agents by ASL while 1) GRM is NOT trained, 2) GRM is trained with self-generated data, and 3) GRM is trained with real data in the third iteration.

Figure 6: The reward hacking phenomenon will occur in agentic self-learning while GRM is not trained. It can be effectively mitigate through training GRM with self-generated data or real data.

That said, even with GRM training we observe gradual convergence after several iterations, suggesting a capacity ceiling governed by the GRM's verification ability. To test whether strengthening the GRM can lift this ceiling, we augment the GRM training in the third iteration with a small amount of real verification data (1% of the Search-R1 training data, constructed through the same method in §4.2). This targeted calibration further improves PM's downstream solving performance and raises the test accuracy compared to training the GRM solely on self-generated data, as reflected by the right-hand curves in Figure 6b.

These findings indicate that ASL's upper bound is effectively determined by the GRM's capability. Self-learning can progressively push the GRM's boundary by exposing it to the current distribution of prompts and solutions, yielding longer and more productive training. Moreover, injecting a modest amount of real verification data in later stages provides a strong anchor for the GRM, refreshes the system's ceiling, and unlocks additional gains. In practice, this suggests a two-phase strategy: rely on self-generated data to continually calibrate the GRM during most of training, then apply a small, carefully curated real-data boost to the GRM to extend ASL's effectiveness in the final rounds.

## 5 CONCLUSION

**Contributions.** In this work, we present Agentic Self-Learning (ASL), a novel multi-role, closed-loop reinforcement learning framework for training LLM-based agents without human supervision or pre-defined rule-based rewards. By unifying task generation (PG), solution synthesis (PM), and adaptive evaluation (Generative Reawrd Model) within a shared backbone and tool environment, ASL enables these components to co-evolve through iterative reinforcement signals. Our controlled studies reveal two pivotal factors for scalable agentic RL: the source of reward signals and the scale of agent-generated task data. ASL addresses both by (i) replacing brittle rule-based judges with a co-trained GRM capable of nuanced verification, and (ii) continuously expanding the training distribution through adaptive task generation informed by entropy-based difficulty control. Empirical results across diverse open-domain QA benchmarks demonstrate that ASL not only surpasses recent self-evolving agents (Search-R1, Absolute Zero, R-Zero) in long-term improvement but also avoids common reward-hacking pitfalls through GRM co-training. Moreover, we identify that GRM can be the bottleneck of the long-time evolving of ASL framework, and injecting small amounts of high-quality real verification data into late-stage GRM training further lifts the performance ceiling, suggesting a practical and scalable hybrid strategy.

**Limitations and Outlook.** Looking forward, the ASL framework provides a general recipe for autonomous agent development, supporting open-ended reasoning, adaptive curricula, and robust reward modeling. However, our current studies are limited to the scenario of text-based search agent. Future work could extend ASL to multimodal agents, multi-turn planning domains, and real-world interactive platforms, paving the way for truly self-improving, domain-agnostic AI systems.

ETHICS STATEMENT

This paper proposes the Agentic Self-Learning framework. All experiments are conducted on publicly available datasets. Thus there is no data privacy concern. Meanwhile, this paper does not involve human annotations, and there are no related ethical concerns.

REPRODUCIBILITY STATEMENT

The data and code of this paper are released at https://anonymous.4open.science/r/Towards-Agentic-Self-Learning-4D63

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

> ### System Prompt for Generative Reward Model and Policy Model
>
> You are Qwen, created by Alibaba Cloud. You are a helpful assistant.
> ## Tools
> You are provided with function signatures within <tools></tools> tags:
> <tools>
> Name:retrieve
> Description: Retrieve relevant information from the locally deployed knowledge base based on the provided list of search terms.
> Input:'query': 'Optional/Required': 'required', 'Parameter Description': 'search terms', 'Parameter Type': 'str'
> </tools>
> For each function call, you should call and then include the json format inputs within <tool_call></tool_call> tags, for example:
> <tool_call>
> {
> "name": tool['name'],
> "arguments": tool['arguments']
> }
> </tool_call>
> For each function call, the result will be returned in the <tool_response></tool_response> tags.
> ## Formats
> Your output should be a combination of the following formats:
> 1. <think>your reasoning thoughts</think>
> 2. <tool_call>
> {
> "name": "retrieve",
> "arguments": {"query": "Beijing cuisine"}
> }
> </tool_call>
> 3. <answer>YOUR ANSWER</answer>
> ## Tasks
> Answer the user's question.
> You can use <think></think> and <tool_call></tool_call> as many times as you want, but you must use <answer></answer> once and only once at the end of your response.
> Your answer should be concise and to the point, without detailed illustrations. For example, <answer>Beijing</answer>.

Figure 7: System prompt for the Generative Reward Model and Policy Model.

**System Prompt for Prompt Generator**

You are Qwen, created by Alibaba Cloud. You are a helpful assistant.
## Tools
You are provided with function signatures within <tools></tools> tags:
<tools>
Name:retrieve
Description: Retrieve relevant information from the locally deployed knowledge base based on the provided list of search terms.
Input:'query': 'Optional/Required': 'required', 'Parameter Description': 'search terms', 'Parameter Type': 'str'
</tools>
For each function call, you should call and then include the json format inputs within <tool_call></tool_call> tags, for example:
<tool_call>
{
"name": tool['name'],
"arguments": tool['arguments']
}
</tool_call>
For each function call, the result will be returned in the <tool_response></tool_response> tags.
## Tasks
generate a question and answer that satisfying the user's request.
You can use <think></think> and <tool_call></tool_call> as many times as you want, but you must use <question></question> and <answer></answer> once and only once at the end of your response.
Your question should be about factual knowledge and can be answered with ONLY ONE concrete entity.
Your answer should be concise and to the point, without detailed illustrations.
For example:
<question>What is the capital of China?</question>
<answer>Beijing</answer>
## Formats
Your output should be a combination of the following formats:
1. <think>your reasoning thoughts</think>
2. <tool_call>
{
"name": "retrieve",
"arguments": {"query": "Beijing cuisine"}
}
</tool_call>
3. <question>YOUR GENERATED QUESTION</question>
4. <answer>YOUR ANSWER</answer>

Figure 8: System prompt for the Prompt Generator.

> **User Prompt for Prompt Generator At Iteration 0**
>
> Please generate a question about {topic}.
> The question should be about factual knowledge and can be answered with ONLY ONE concrete entity.
> Use the tool to help you generate the question, ensure that the question is solvable and the tool is necessary to answer the question.
> Output with the following format:
> <question>
> ...a generated question...
> </question>
> <answer>
> ...the answer to the generated question...
> </answer>

Figure 9: User prompt for Prompt Generator tt iteration 0.

> **User Prompt for Prompt Generator At Iteration 1**
>
> I asked a question to my students:
> {question_prompt}
>
> The true answer to this question is:
> {answer_to_question}
>
> Most student {answer_type}LY answer this question, therefore, to better train my students, I want you to generate a {question_type} question based on the current question.
> You should generate the new question that is similar to the current question but requires new knowledge and is {question_type} to answer.
> Follow the following instructions to generate the new question:
> 1. The new question should be about factual knowledge and can be answered with ONLY ONE concrete entity.
> 2. Use the tool to help you generate the new question.
> 3. The question must be solvable and the tool must be necessary to answer the question. Your final output should be of the following formats:
> <question>
> ...a new question...
> </question>
> <answer>
> ...the answer to the new question...
> </answer>

Figure 10: User prompt for Prompt Generator at iteration 1.

> **User Prompt for Prompt Generator At Later Iterations**
>
> I asked a question to my students:
> {question_prompt}
>
> The true answer to this question is:
> {answer_to_question}
>
> Most student {answer_type}LY answer this question, therefore, to better train my students, I
> want you to generate a {question_type} question based on the current question.
> You should use the tool to retrieve the relevant infomation about the question, then generate
> the new question by {question_constraint} one reasoning hop to the current question.
> e.g. {demo}
> The new question should be about factual knowledge and can be answered with ONLY ONE
> concreate entity.
> Output with the following format:
> <question>
> ...a new question...
> </question>
> <answer>
> ...the answer to the new question...
> </answer>

Figure 11: User Prompt for Prompt Generator At Later Iterations

> **User Prompt for Generative Reward Model**
>
> I asked a question to my student:
> {question_prompt}
>
> My student's answer is:
> {completion}
>
> I want you to help me verify the correctness of my student's answer.
> Use the tool and thinking to analyze, then draw your conclusion in the following format:
> <answer>
> conclusion: correct/wrong
> </answer>

Figure 12: User prompt for Prompt Generator at later iterations.

---

**Example of Prompt Generator**

<think>
To generate a harder question, I first need to retrieve information about baharat and Tavuk göğsü recipe to understand the context. Then I will add one reasoning hop by identifying a related ingredient or concept.
</think>
<tool_call>
{ "name": "retrieve", "arguments": { "query": "baharat Tavuk göğsü" } }
</tool_call>
<|im_start|>user
<tool_response>

id: 3686413
content:
"Tavuk göğsü"
Tavuk göğsü Tavuk göğsü (, , ""chicken breast"") is a Turkish milk pudding made with shredded chicken breast. It became one of the most famous delicacies served to the Ottoman sultans in the Topkapı Palace. It is today considered a signature dish of Turkey. The traditional version uses white chicken breast meat, preferably freshly slaughtered capon. The meat is softened by boiling and separated into very fine fibers or ""rillettes"". Modern recipes often pound the meat into a fine powder instead. The meat is mixed with milk, sugar, cracked rice and/or other thickeners, and often some sort of flavoring such

id: 3686414
content:
"Tavuk göğsü"
as cinnamon. The result is a thick pudding often shaped for presentation. The dish is very similar to the medieval ""white dish"", ""blancmange"" that was common in the upper-class cuisine of Europe, and mentioned in ""The Canterbury Tales"" (though blancmange has since evolved into very different forms in modern Europe and Latin America). Tavuk göğsü Tavuk göğsü (, , ""chicken breast"") is a Turkish milk pudding made with shredded chicken breast. It became one of the most famous delicacies served to the Ottoman sultans in the Topkapı Palace. It is today considered a signature dish of Turkey. The traditional version

id: 5389908
content:
"Göksu, Mut"
Göksu, Mut Göksu is a town in Mersin Province, Turkey. Göksu at is a part of Mut district of Mersin Province. It is from Mut and from Mersin. The population is 2146 as of 2012 The road to the town diverges from the Mersin-Karaman highway at around the Alahan Monastery. (But while the monastery is to the east, the town is to the west) The town is established in 1995 by merging four neighboring villages, namely Kravga, Bayır, Esen and Köprübaşı.

</tool_response><|im_end|>
<|im_start|>assistant
# <think> and <tool_call> for 5 more rounds
<think>
From the retrieved information, I can identify that baharat typically includes spices like cinnamon, cloves, and black pepper. Since cinnamon is a common ingredient in Tavuk göğsü, a harder question could be about a baharat spice that is not commonly found in Tavuk göğsü.
</think>
<question> What is a common ingredient in baharat that is not typically found in traditional Tavuk göğsü? </question>
<answer> mace </answer>

Figure 13: Example of Prompt Generator.

**Example of Policy Model**

<think>
Let's retrieve information about solo artists known for their "Beatlesque" style to find the answer.
</think>
<tool_call> { "name": "retrieve", "arguments": { "query": "Beatlesque solo artist" } }
</tool_call>
<|im_start|>user
<tool_response>

id: 196567
content:
Beatlesque
Beatlesque ""Beatlesque"" () or ""Beatles-esque"" describes a musical resemblance to the English rock band the Beatles. The term's definition lacks consensus and it has been applied inconsistently to a wide number of disparate artists. To better explain what the word might mean, eight possible answers were formulated by radio producer Kevin Howlett, music professor Rob Bowman, and Klaatu drummer Terry Draper: ""Toronto Star""s Jack Sakamoto commented: ""[some people's] notion of that sound includes everyone from Panic! at the Disco to Billy Joel to the Red Hot Chili Peppers.

id: 196568
content:
Beatlesque
for the adjective their music has spawned."" Notable acts described as ""Beatlesque"" Other Beatlesque ""Beatlesque"" () or ""Beatles-esque"" describes a musical resemblance to the English rock band the Beatles. The term's definition lacks consensus and it has been applied inconsistently to a wide number of disparate artists. To better explain what the word might mean, eight possible answers were formulated by radio producer Kevin Howlett

id: 14870332
content:
"George Harrison"
George Harrison George Harrison (25 February 1943 – 29 November 2001) was an English musician, singer-songwriter, music and film producer who achieved international fame as the lead guitarist of the Beatles. Often referred to as ""the quiet Beatle"", Harrison embraced Indian culture and helped broaden the scope of popular music through his incorporation of Indian instrumentation and Hindu-aligned spirituality in the Beatles' work.

</tool_response><|im_end|>
<|im_start|>assistant
# <think> and <tool_call> for 2 more rounds
<think>
From the information retrieved, George Harrison's post-Beatles career continued to be "Beatlesque". He maintained a connection to the Beatles sound and style in his solo work, including with the band Traveling Wilburys. However, to directly answer the question about a solo artist known for his "Beatlesque" career, let's conclude with the most relevant find.
</think>
<answer>
George Harrison, often known as "the quiet Beatle," was a significant figure who continued to be "Beatlesque" in his solo career, especially through his work with the Traveling Wilburys. However, if we are looking for a solo artist who is specifically referenced as "Beatlesque," notable acts such as Billy Joel, who has been described as "Beatlesque," could fit the description based on various interpretations of the term. Nonetheless, George Harrison is a prominent example of an artist who maintained and continued the "Beatlesque" legacy in his solo work.
</answer>

Figure 14: Example of Policy Model.

> **Example of Generative Reward Model**
>
> <think>
> To verify the student's answer, we need to check if "Pinko Cronkite" was indeed the nickname Archie Bunker used and if it was inspired by "Sklanton" or "Slickdan" from Popeye's comic strip.
> </think>
> <tool_call> { "name": "retrieve", "arguments": { "query": "Archie Bunker Pinko Cronkite All in the Family" } } </tool_call>
> <|im_start|>user
> <tool_response>
>
> id: 12892536
> content:
> "Walter Cronkite"
> popular situation comedy ""All in the Family"", during which the lead character Archie Bunker would sometimes complain about the newsman, calling him ""Pinko Cronkite."" Cronkite trained himself to speak at a rate of 124 words per minute in his newscasts, so that viewers could clearly understand him. In contrast, Americans average about 165 words per minute, and fast, difficult-to-understand talkers speak close to 200 words per minute. In 1968, the faculty of the E. W. Scripps School of Journalism at Ohio University voted to award Cronkite the Carr Van Anda Award ""for enduring contributions to journalism."" In 1970, Cronkite received
>
> id: 864474
> content:
> "Archie Bunker"
> Archie Bunker Archibald ""Archie"" Bunker is a fictional character from the 1970s American television sitcom ""All in the Family"" and its spin-off ""Archie Bunker's Place"", played by Carroll O'Connor. Bunker, a main character of the series, is a World War II veteran, blue-collar worker, and family man. Described as a ""lovable bigot"", he was first seen by the American public when ""All in the Family"" premiered on January 12, 1971, where he was depicted as the head of the Bunker family.
>
> id: 15975305
> content:
> "All in the Family"
> All in the Family All in the Family is an American sitcom TV-series that was originally broadcast on the CBS television network for nine seasons, from January 12, 1971 to April 8, 1979. The following September, it was continued with the spin-off series"" Archie Bunker's Place"", which picked up where ""All in the Family"" had ended and ran for four more seasons. ""All in the Family"" was produced by Norman Lear and Bud Yorkin. It starred Carroll O'Connor, Jean Stapleton, Sally Struthers, and Rob Reiner.
>
> </tool_response><|im_end|>
> <|im_start|>assistant
> # <think> and <tool_call> for 1 more rounds
> <think>
> The retrieved information does not provide any details about the inspiration for the nickname "Pinko Cronkite" being derived from "Sklanton" or "Slickdan" from Popeye's comic strip. Therefore, the student's answer seems to contain inaccurate information regarding the origin of the nickname.
> </think>
> <answer>
> conclusion: wrong
> </answer>

Figure 15: Example of Generative Reward Model.

> **Example of Evolved Question**
>
> **Iteration 1:**
>
> **Q:** Which Israeli city is known as the holiest city in Judaism?
> **A:** Jerusalem.
>
> **Iteration 2 (harder—depth):**
>
> **Q:** Which most important religious site is located in the holiest city in Judaism?
> **A:** The Western Wall.
>
> **Iteration 3 (harder—breadth):**
>
> **Q:** What is the name of the most expensive hotel in the city that contains the Western Wall, located in the religious capital of Israel, which is surrounded by the claimed capital of the State of Palestine?
> **A:** Park Hyatt Jerusalem.

Figure 16: Example of Evolved Question. ASL indeed produces increasingly difficult, realistic, and diverse questions during multi-role iterations

# A    PROMPTS

# B    GENERATION EXAMPLES OF THREE ROLES IN ASL

# C    EVOLUTION OF QUESTION

# D    DETAIL OF ASL SETTINGS

The Prompt Generator training phase terminates when the stored Policy Model training data $D_{PM}^{(t)}$ reach a specified amount, and we set it to 2000 in the iteration 0 and 4000 in later iterations.

The Generative Reward Model and Polcy Model training phase terminates when their training reward curve slope reach a minimum threshold 0.05.

The batchsize $N$ is set to 128, $K$ is set to 5, and $M$ is set to 10. The threshold $C$ is set to 0.8. The seed of the whole experiment is set to 37.

# E    THE USE OF LARGE LANGUAGE MODELS

We used LLMs to assist with part of the translation and language polishing during the preparation of this paper.

