# OpenReview forum: "Towards Agentic Self-Learning LLMs in Search Environment"
_ICLR.cc/2026/Conference — Submitted to ICLR 2026_

### Official Review · Reviewer_JUBh · 2025-10-28

**Soundness:** 2
**Presentation:** 3
**Contribution:** 2
**Rating:** 2
**Confidence:** 4

**Summary:**

This paper proposes a self-learning paradigm for improving the capabilities of LLM based agents. The authors propose a method that leverages GRMs, which they find to outperform the rule based methods. The benefit of the GRM is that the GRM can be further improved through RL training as the prompt generator produces harder and harder prompts, the GRM can be further trained to produce rewards for this task.

**Strengths:**

This area of self-learning is of significant importance for producing autonomous agents that can improve their performance for new tasks. As far as this reviewer can tell, the work is original and distinct from the previous works in the literature.

**Weaknesses:**

1. Results seem to be across a single seed of training (I couldn't find information about training setup i.e. number of seeds in the manuscript) which is very limited for research in RL. Even a small handful of seeds (i.e. 5) combined with the correct statistical methods (i.e. IQM[1]) can provide valuable insights. I strongly suggest that the authors work to include additional training runs of their method

2. Combined with #1, results are shown as point estimates and therefore we cannot evaluate the statistical significance of the results. Leveraging IQM [1] would provide insights into whether the experimental results are significantly different from each other or not.

3. The figures are hard to read as the text and labels are much too small.

4. The authors re-introduce acronyms for their methods (i.e. Generative Reward Model (GRM) line 283 and line 068) throughout the text. Please introduce the acronym once when the method is first introduced, and then use the acronym going forward.

5. I think that a discussion of scaling in RL is needed to give the reader full context of the literature. From the way the paper is written, a reader could think that scaling isn't done in RL outside of this proposed setting which is untrue. There are many works [2-4] that study parameter scaling in RL, while a few works [5, 6] provide similar insights into the relationship between scaling and the amount of data used during training, albeit in a more traditional RL problem setting.

Citations
1. https://arxiv.org/abs/2108.13264
2. https://arxiv.org/pdf/2506.17204
3. https://arxiv.org/abs/2405.16158
4. https://openreview.net/forum?id=kfYxyvCYQ4
5. https://arxiv.org/pdf/2506.03404?
6. https://arxiv.org/abs/2503.05126

**Questions:**

1. In the GRM training phase, the GRM performs N independent rollouts that generate a series of predictions for the score. These predicted scores are then compared against the reference scores - where do the reference scores come from? This seems like a bottleneck that the authors attempted to avoid by using the GRM but if GRM training relies on an outside verifier (i.e. human labels), then the positives of the GRM are lost.
2. How could this method be translated into more traditional RL settings (i.e. interactions with Atari or Mujoco environments)?

---

> ### Author Response · Authors · 2025-11-19
>
> ## Thank you very much for reviewing our paper. Below are our responses to your comments.
>
> ### Regarding strengths
> + Thank you very much for recognizing the research topic and the novelty of our methods!
>
> ### Regarding weaknesses
> + W1: Thank you very much for your valuable suggestions!
>     - Following your advice, we will further supplement our results by running with multiple random seeds to better demonstrate the significant advantages of our method over other self-learning approaches, as well as its ability to match the performance of Search-R1 [1] without human data. Since Agentic RL training is time-consuming, we will submit the experimental results at a later date. Thank you again for your suggestion.
>     - As Agentic RL indeed requires substantial training, related works such as Search-R1 [1] and R1-searcher [2] have also adopted similar experimental settings.
> + W2: Thank you for your detailed suggestions for improvement! As noted above, we will submit the experimental results at a later date. Thank you again for your suggestions.
> + W3: Thank you for your suggestion! We will adjust the presentation of images across groups in the revised version to improve the reading experience for readers.
> + W4: Thank you for your suggestion! We will follow your recommendation in the revised version to introduce acronyms initially and subsequently use the corresponding acronyms consistently.
> + W5: Thank you very much for your valuable suggestion! In Section 3.1, under the Agentic RL setting for the Deep Search task, we intend to emphasize the importance of data scaling (and reward signals) to motivate the design of our ASL framework; we do not wish to mislead readers into thinking that there is no research on RL scaling beyond this setting, and we apologize for the lack of clarity in the corresponding logic. Thank you for providing the references; these are indeed excellent existing works on RL scaling. Based on your suggestion, we will add a discussion of prior work before Section 3.1 in the revised version, and on that basis present our research setting and findings. Thank you again for your valuable suggestion!
>
> ### Regarding questions
> + Q1: Thank you very much for your question. The training data for the Generative Reward Model comes from the Prompt Generator training phase, specifically the (x, y, s) mentioned in line 270, where x and y are the question and an answer respectively, and s is the correctness score of that answer, i.e., the reference score mentioned in line 279. The GRM training process performs RLVR training based on such (x, y, s) data: for each (x, y) pair, the GRM analyzes and determines whether the reply correctly answers the question, produces a judgment score ŝ, compares ŝ with the true correctness score s, and obtains a 0/1 score indicating whether the GRM’s judgment is correct; this score is the final reward the GRM receives. We apologize again for any confusion, and in the revised version we will more clearly describe the GRM’s RLVR training process by referencing the earlier section.
> + Q2: You raised a very valuable question! Since our proposed ASL method makes no assumptions about tool types and does not incorporate any properties unique to the Search environment into the framework design, we believe ASL can be generalized to more agentic task scenarios. For example, one of the baselines compared in this paper, R-zero [3], is validated in the math reasoning setting, and Absolute Zero [4] is validated in the code setting. Thank you very much for your valuable suggestion. We will extend ASL to more agentic task scenarios, such as Atari and Mujoco and other traditional RL settings, as one of our important future goals.
>
> ### Summary
> + Once again, we sincerely thank you for your meticulous review and thoughtful, insightful suggestions. They greatly help improve the presentation of our methods and experiments and the clarity of the writing. We would be very happy to continue our discussion with you!
>
> [1] Jin B, Zeng H, Yue Z, et al. Search-r1: Training llms to reason and leverage search engines with reinforcement learning[J]. arXiv preprint arXiv:2503.09516, 2025.
>
> [2] Song H, Jiang J, Min Y, et al. R1-searcher: Incentivizing the search capability in llms via reinforcement learning[J]. arXiv preprint arXiv:2503.05592, 2025.
>
> [3] Huang C, Yu W, Wang X, et al. R-Zero: Self-Evolving Reasoning LLM from Zero Data[J]. arXiv preprint arXiv:2508.05004, 2025.
>
> [4] Zhao A, Wu Y, Yue Y, et al. Absolute zero: Reinforced self-play reasoning with zero data[J]. arXiv preprint arXiv:2505.03335, 2025.

---

> ### Author Response · Authors · 2025-11-24
>
> Thank you once again for reviewing our paper and for the time and effort you devoted! We have uploaded the revised manuscript via the Rebuttal Revision. We have revised many expressions throughout the manuscript based on your comments. Thank you for your efforts!

---

> ### Author Response · Authors · 2025-11-27
>
> Dear Reviewer JUBh, thank you again for your time and effort in reviewing our manuscript. We hope our responses have adequately addressed your concerns. We are happy to engage in further discussion if you have any additional questions.

---

### Official Review · Reviewer_WoRw · 2025-10-29

**Soundness:** 3
**Presentation:** 3
**Contribution:** 3
**Rating:** 6
**Confidence:** 3

**Summary:**

The work introduces a new framework, Agentic Self-Learning, that aims to train an LLM with minimal human effort. This framework comprises three components: a prompt generator that creates tasks for a policy model to solve, and a generative reward model trained against a reference score to provide a reward signal to the prompt generator, which maximizes the entropy of the return, and to the policy model, which maximizes the return. Agentic Self-Learning is evaluated against competitive baselines, and multiple ablations are presented to understand its behavior.

**Strengths:**

1. The presented method is novel and creative, focusing on a recent topic of artificial intelligence.

2. The reading flow is good. The method is clearly presented.

**Weaknesses:**

I. Many details are missing to assess the presented method properly:

   a. By learning to increase the entropy of the return over a batch of predicted tasks, the prompt generator is incentivized to make simple tasks simpler and hard tasks harder. A comment that provides intuition for why this behavior leads to tasks well-adapted to the current policy model is missing.

   b. No details are provided to explain the way the difficulty feedback is implemented. Given that the prompt generator is trained to maximize entropy, listening to the feedback would increase the loss because it would guide the model to generate hard tasks simpler and simple tasks harder. Clarifying this aspect would be helpful.

   c. Line 280, the statement "This comparison produces a binary correctness indicator for each rollout" is not specific enough. Moreover, in Line 279, "we compare it against the reference score s", no details are given to describe the reference score.

   d. In Section 3.1, the environment in which the proposed method is evaluated is not described. Adding a short description would be beneficial.

   e. Line 312, "all roles share parameters", no information on which parameters are shared is disclosed.

   f. VeRL is used as a base algorithm; however, no justification for this choice is provided. Additionally, it would be preferable to discuss how the performances would change if another algorithm were selected.

   g. The lack of a preliminary Section increases the entry barriers of the presented approach, which lowers the potential audience for this work. Adding a preliminary section would strengthen the impact of the presented work.

   h. Adding a comment on the importance of the hyperparameters such as the length of the different phases would improve the usefulness of the presented approach. Additionally, commenting on the influence of the capability of the initial policy on the performance of the presented framework would be interesting.


II. Overall, the presentation can be improved:

   a. In Figure 5, the confidence intervals can be added.

   b. Parentheses are missing around many citations. For example, Line 139, "Shi et al." should have parentheses. Moreover, many citations do not have dates.

   c. The acronym PURM, in Line 98 is not defined.

   d. In Line 188, "(1)" should be written instead of the first "(3.1)", and "(2)" should be written instead of the second "(3.1)".

   e. The font size of the all figures should be much bigger.

**Questions:**

N/A

---

> ### Author Response · Authors · 2025-11-18
>
> ## Thank you very much for reviewing our paper. Below are our responses to your comments.
>
> ### Regarding strengths
> + Thank you very much for recognizing the research topic, the proposed method, and the writing of the paper!
>
> ### Regarding weaknesses
> + W1: We sincerely appreciate your detailed evaluation and suggestions!
>     - a. We apologize for causing your misunderstanding. Our reward assignment for the Prompt Generator is not “the entropy of the return over a batch of predicted tasks,” but “the entropy of the score distribution after a single task is answered multiple times” (lines 266–269). If we regard a question as an exam paper, then the score distribution from multiple rollouts can be seen as a class’s grade distribution under that exam; its entropy effectively reflects how well this exam (this question) distinguishes students’ abilities. Therefore, optimizing this reward value will make “hard problems become easier, and easy problems become harder,” which yields questions that help the Policy Model observe rich reward signals and thus learn effective problem-solving strategies. We again apologize for the confusion, and we will describe the optimization objective of the Prompt Generator more clearly in the revised version.
>     - b. According to the description in A, the training objective of the Prompt Generator is indeed to make “hard problems become easier, and easy problems become harder.” In lines 296–302, we further construct the prompt data needed for the next round of Prompt Generator training via the Policy Model’s Difficulty Feedback. We apologize for the confusion, and we will, in the revised version, combine point A to more clearly and logically describe the data construction mechanism for Prompt Generator training.
>     - c. We apologize for the difficulty in understanding. Here, the training data for the Generative Reward Model is in the form (x, y, s) mentioned in line 270, where x and y are the question and an answer respectively, and s is the correctness score of that answer, i.e., the reference score mentioned in line 279. The GRM’s training process is RLVR based on such (x, y, s) data: for each (x, y), it analyzes and judges whether the reply correctly answers the question, produces a discriminative score s_hat, compares s_hat with the true correctness score s, and obtains a 0/1 score representing “whether the GRM judged correctly”; that score is the final reward the GRM receives. We again apologize for the confusion, and we will clarify the GRM’s RLVR training process by referencing the above in the revised version.
>     - d. Thank you very much for your suggestion! Because we wanted to quickly highlight the motivation brought by these experiments, we placed the introduction to the experimental setup later in Section 4.1 (briefly mentioned in one sentence at line 185). Following your suggestion, we will add a brief introduction to the experimental setup here to help readers better understand the background and conclusions of the experiments in Section 3.1.
>     - e. Thank you very much for your suggestion! In the ASL framework, we let the three roles share all parameters of the entire model, which allows the learning of three different tasks to fully interact and generalize, achieving collaborative improvement. Thank you for your suggestion; we will add a description of how the three roles share the model in the revised version.
>     - f. VeRL is a recently popular and easy-to-use RL training framework. To best realize ASL’s design and functionality, we implemented the code based on this framework. Because our method does not make any assumptions about the RL algorithm and focuses on the design of a multi-role agent system, we believe that replacing the RL algorithm (e.g., switching to GRPO or reinforce++) will not affect the performance of the ASL framework at a macro level.
>     - g. Thank you very much for your valuable suggestion! Due to space limitations, we indeed lack a preliminary section introducing the overall task and domain background. We will add a preliminary section in the revised version to introduce the corresponding basic tasks and notation. Thank you again for your valuable suggestion!

---

> > ### Author Response · Authors · 2025-11-18
> >
> > + W1 (h)
> >    - h. You raised a very valuable question! Yes, we have already considered this in the design of the ASL framework and reduced total computational overhead by limiting the training phase lengths of the Prompt Generator and the Generative Reward Model respectively (because we found that beyond a certain amount, training these two has little impact on the policy model performance we need). In practice, we control early stopping by setting a maximum number of questions generated by the Prompt Generator and a minimum reward slope for training the Generative Reward Model. Additionally, we believe that a stronger initial policy is more conducive to the ASL framework learning the collaborative improvement of the three tasks, thereby improving the training efficiency and performance ceiling of the whole ASL framework. Thank you very much for your suggestion; we will add these details and comment in the revised version to improve the usefulness of the presented approach.
> > + W2: Thank you for your detailed suggestions for improving this paper!
> >     - a. Thank you for your suggestion! In the revised version, we will add confidence intervals to Figure 5 to better reflect the synergistic improvement of the capabilities of the three roles.
> >     - b. Thank you for your suggestion! We will check all citations to ensure they are appropriate to the context and free of formatting errors.
> >     - c. We are very sorry for the confusion this caused in your reading! This was a typo, and we will correct it in the revised version.
> >     - d. Thank you very much for your suggestion! It indeed should not refer to the entire section title; we will follow your advice and change it to (1) and (2), corresponding to the two paragraphs below.
> >     - e. Thank you very much for your suggestion! In the revised version, we will adjust the presentation of each group of images to provide readers with a better reading experience.
> >
> > ### Summary
> > + Once again, we sincerely thank you for your meticulous review and your thoughtful and insightful suggestions. They are immensely helpful for the presentation of our methods and experiments and for improving the clarity of the writing. We would be very happy to continue the discussion with you!

---

> ### Author Response · Authors · 2025-11-24
>
> Thank you once again for reviewing our paper and for the time and effort you devoted! We have uploaded the revised manuscript via the Rebuttal Revision. We have revised many expressions throughout the manuscript based on your comments. Thank you for your efforts!

---

> ### Author Response · Authors · 2025-11-27
>
> Dear Reviewer WoRw, thank you again for your time and effort in reviewing our manuscript. We hope our responses have adequately addressed your concerns. We are happy to engage in further discussion if you have any additional questions.

---

### Official Review · Reviewer_q5p7 · 2025-10-31

**Soundness:** 2
**Presentation:** 2
**Contribution:** 3
**Rating:** 4
**Confidence:** 3

**Summary:**

This paper examines whether self-learning using LLM generated rewards can outperform rule based verifiable rewards.Their agentic self-learning approach outperforms strong baselines such as Search-R1. The primary concern of reward hacking is mitigated by continued training of the Generative Reward Model and late-stage verification data injection. This framework consists of multiple components: the prompt generator, rewarded by the entropy of the score distribution of the downstream policy; the generative reward model, this is trained to mimic the verifiable reward and a policy model, which provides candidate solutions.

**Strengths:**

The paper tackles a challenging problem and proposes a novel framework which has some promise of self-improvement. It identifies the generative reward model is the bottleneck for self improvement, as well as identifying/mitigating an instance of reward hacking. There are interesting design decisions such as sharing of policy and reward model parameters.

**Weaknesses:**

The primary concern with RLAIF is the difficulty in avoiding reward hacking, whilst interesting I am yet to be convinced the proposed approach solves this problem.

Prompt Generator: Whilst rewarding based on the entropy of responses seems like an intuitive solution, it isn't clear how it would avoid reward hacking for solutions with many answers, i.e. always asking to roll a 1000 sided dice just has high entropy in responses/scores but is a degenerate prompting strategy.

Reward Model: As I understand it, this model is trained on the binary signal of if the reward model matches the verifiable reward output. Therefore, I am unsure how it can outperform the verifiable reward signal it is approximating when evaluated on the same task (training a model of the verification signal could perhaps generalize to domains where a signal is not present).

Policy: Asking the prompt generator for easier or harder prompts based on the score initially seems sensible however it could induce failure modes induced by the policy being punished for doing better, as if it answers correctly the questions become harder. Whilst perhaps not breaking it warrants discussion, particularly regarding the efficiency of learning.

Figure text is too small, with no information regarding seeds or uncertainty.

4.4 Ablation is useful to see that each component is learning however, as it's a closed loop system, it's hard to tell if the model has reward hacked without witnessing model outputs. For instance, I'm struggling to infer how you would notice the reward hacking behavior you mention from this plot. Therefore we're just taking the claim that at least one reward hack exists at face value without knowledge of how you discovered it and if there are any other degeneracies.

**Questions:**

1. How does continual training mitigate reward hacking? What details can you provide toward the exact training mechanism?
2. If injecting verification data seems to help, is this not contrary to the hypothesis that GRM data is better?
3. Fig 2, why the sudden drop in reward for the rule-based method at the end?
4. Is there a tradeoff between learning efficiency and final performance?
5. How did you discover the reward hacking behavior in 4.4? How can we be assured there aren't other failure modes?
6. You mention the generative judge "can provide more informative and tolerant reward signals", are you able to elaborate? Which aspect of training induces these qualities

---

> ### Author Response · Authors · 2025-11-18
>
> ## Thank you very much for reviewing our paper. Below are our responses to your comments.
>
> ### Regarding strengths
> + Thank you very much for recognizing the importance of the research topic and the novelty and effectiveness of our method!
>
> ### Regarding weaknesses
> + W1: About reward hacking in RLAIF
>     - Prompt Generator: Your idea is very reasonable! This is exactly what we considered: the Prompt Generator can induce the GRM to assign high-entropy scores by generating random/unsolvable questions, thereby cheating for rewards. To address this problem, we added a question verification mechanism (lines 261–265): we do not fully trust the questions generated by the Prompt Generator; instead, we submit its questions and answers to the GRM for verification. If consistent correct judgments cannot be obtained (in our cases, with more than 80% accuracy), we consider the question to be at risk of hacking (containing randomness or exceeding the GRM’s discriminative ability) and discard it, not adding it to the training data for the Policy Model and Generative Reward Model in that round.
>     - Reward Model: This is a very valuable question! In our early validation experiments we noticed that rule-based reward functions perform worse than a GRM with good discriminative ability in the Deep Search setting (Figure 2). This is because while the GRM has good discriminative ability, it also has stronger generalization. For example, we observed a question asking in which century Napoleon crowned himself emperor; the labeled answer is 19, and when the model answers “nineteenth century,” the rule-based reward function assigns a score of 0, leading to the loss of such high-quality trajectory signals.
>     - Policy Model: Because the training of our roles is staged relative to each other, the Policy Model will not realize the latent mechanism in ASL that “better answers lead to harder questions,” and will only be influenced by the question–reward of the current round to pursue stronger question-answering ability. (Side note: Perhaps humans exhibit similar behavior, i.e., a child might pretend to be clumsy to avoid work, but this is because the child has long-term context and still remembers the previous round’s answer when answering the next round’s question; whereas our Policy Model does not have the previous round’s question as context when trained on the next round’s question.)
> + W2: Thank you very much for your suggestion. We will further refine the visual appearance of the figures in the revised version and add information regarding experimental seeds in the text.
> + W3: We apologize for the confusion caused during reading! In Section 4.4, our determination of reward hacking is not based on Figure 5 (which corresponds to the conclusion of three-role collaborative improvement in Section 4.3), but is based on Figure 6(a). From Figure 6(a), we can see that although the Prompt Generator continues to obtain increasingly higher rewards during its training phase, the Policy Model’s reward during training and its accuracy on the test set no longer improve. This indicates that the Prompt Generator has exploited the limitations of the Generative Reward Model’s capability and started generating questions that are difficult to judge correctly to cheat for rewards. Again, we sincerely apologize for the confusion caused by our improper layout!

---

> > ### Author Response · Authors · 2025-11-18
> >
> > ### Regarding questions:
> > + Q1: In Section 4.4 we first conducted an ablation on ASL without training the GRM and discovered a potential reward hacking phenomenon, i.e., the Prompt Generator hacks the GRM’s limited capability to cheat for rewards (lines 394–425). The main mechanism lies in generating questions that are too difficult (relative to the GRM’s capability) or even unsolvable (relative to the information provided by the tool environment), causing the GRM to be unable to judge answer correctness and thus produce near-random scoring behavior, resulting in a high-entropy score distribution (lines 414–416). Subsequently, we primarily relied on the complete ASL method, i.e., adding a GRM training phase in each iteration to continuously raise the GRM’s capability ceiling, thereby penalizing the Prompt Generator’s strategy of generating such overly difficult questions and mitigating the reward hacking phenomenon (lines 426–452). Furthermore, we found that introducing a small amount of real data to the GRM’s training in the later stages can further raise the ceiling of the entire ASL framework (lines 452–457). These discussions helped us understand and effectively mitigate the reward hacking phenomenon.
> > + Q2: You raised a very interesting question! We believe this is not contradictory: In Section 3.1(1), we experimentally verified that the GRM as a reward signal can guide the policy model’s effective training better than a rule-based reward signal based on training data answers; but this does not mean that the GRM would become worse when trained on discriminative data constructed from training data. The advantage of the GRM is that it can extend rigid rule-based scores to softer discriminative boundaries and will not judge a reasoning trajectory as an incorrect answer due to a spelling or paraphrase issue in the final answer, which helps the policy model learn more good strategies. However, after extensive training on synthetic data within the ASL framework, the GRM gradually loses realism; at that point, introducing a small amount of real data for training can effectively alleviate this.
> > + Q3: We believe this is an overfitting phenomenon occurring in Search-r1 on the training set, leading to degraded performance on the test data. Methods based entirely on synthetic data (R-zero, Absolute Zero, ASL) do not exhibit this phenomenon early on because they ensure sufficient diversity in the training data.
> > + Q4: You raised a very valuable question! Yes, we have already considered this when designing the ASL framework and reduced total computational overhead by limiting the individual training phase lengths of the Prompt Generator and Generative Reward Model (because we found that beyond a certain amount, the training of these two has little impact on the policy model performance we need). In practice, we set a maximum number of questions generated by the Prompt Generator and a minimum reward slope threshold for the Generative Reward Model to control early stopping for both. We will add these details to the revised version to better present how ASL minimizes computational overhead as much as possible. As it stands, ASL achieves performance comparable to the best performance of Search-r1 (at 2800 steps) at around 3800 steps, which means ASL requires less than 50% additional training steps to completely eliminate reliance on labeled data and RLVR and achieve self-learning, which we believe is a very worthwhile cost.
> > + Q5: As shown in Figure 6(a), we observed that during iterative training of the Prompt Generator and Policy Model, although the training reward keeps increasing (the drop at the start of each round relative to the previous round is due to the other role’s relative improvement), the model’s accuracy on test tasks stops changing from the third round. This is a typical manifestation of reward hacking, i.e., the reward on training prompts continues to rise on the surface but does not actually generalize to the test set [1]. In response to this typical manifestation, we further analyzed its underlying principle: due to the GRM’s limited capability, the Prompt Generator can generate questions that are too difficult (relative to the GRM’s capability) or even unsolvable (relative to the information provided by the tool environment), causing the GRM to be unable to judge answer correctness and thus produce near-random scoring behavior, resulting in a high-entropy score distribution.

---

> > > ### Author Response · Authors · 2025-11-18
> > >
> > > + Q6: Thank you very much for your question. Rule-based rewards perform well in game, code, and math scenarios because their answers are often unique and unambiguous. However, in the Deep Search scenario, the same answer may have multiple possible expressions (and some questions may even have multiple possible answers). In this case, using a rule-based reward based solely on the labeled answer in the training data often causes a loss of expressive ability, thereby limiting the improvement space for the model in RL training. A generative judge can effectively solve this problem and provide high-quality reward signals for the policy model’s RL training. For example, we observed a question asking in which century Napoleon crowned himself emperor; the labeled answer is 19, and when the model answers “nineteenth century,” the rule-based reward function assigns 0, resulting in the loss of such excellent trajectory signals, whereas the GRM can correctly assign reward = 1 to that trajectory.
> > >
> > > ### Summary
> > > + Once again, thank you for your thoughtful and insightful suggestions; they greatly help improve the quality of our discussions on the paper’s motivation and results. We would be very happy to continue the conversation with you!
> > >
> > > [1] Skalse J, Howe N, Krasheninnikov D, et al. Defining and characterizing reward gaming[J]. Advances in Neural Information Processing Systems, 2022, 35: 9460-9471.

---

> ### Author Response · Authors · 2025-11-24
>
> Thank you once again for reviewing our paper and for the time and effort you devoted! We have uploaded the revised manuscript via the Rebuttal Revision. In response to your question W2 and Q4, we have presented the detailed experiment settings in Appendix D.

---

> ### Author Response · Authors · 2025-11-27
>
> Dear Reviewer q5p7, thank you again for your time and effort in reviewing our manuscript. We hope our responses have adequately addressed your concerns. We are happy to engage in further discussion if you have any additional questions.

---

### Official Review · Reviewer_UEib · 2025-10-31

**Soundness:** 2
**Presentation:** 4
**Contribution:** 2
**Rating:** 4
**Confidence:** 2

**Summary:**

This paper presents Agentic Self-Learning (ASL). This closed-loop reinforcement learning framework enables LLM agents to self-improve without human-labeled data or rule-based rewards. Through controlled experiments in a search-based agent environment, the authors identify two key determinants of scalable agentic training: the reward source and the scale of the data. ASL unifies task generation, policy execution, and reward evaluation within a shared LLM backbone, allowing the Prompt Generator, Policy Model, and GRM to co-evolve through iterative reinforcement. Empirical results show that ASL achieves sustained performance gains, surpassing strong baselines such as Search-R1, Absolute Zero, and R-Zero, even under zero-labeled-data conditions.

**Strengths:**

- Human-free learning: the authors demonstrate scalable self-learning without human annotations or rule-based rewards, a significant step toward autonomous LLM training/self-improvement.
- Generalizability of the method and experiments: the work offers, in principle, a scalable, domain-agnostic recipe for self-improving agents, extensible to broader open-domain or multimodal tasks.

**Weaknesses:**

- Limited domain scope: the experiments are confined to a text-based search QA environment, leaving it unclear how ASL would generalize to other domains (e.g., coding or multimodal tasks).
- Complex training loop: the multi-role co-evolution process increases implementation complexity and computational cost.
- Non-significant experimental results: most experimental curves do not seem to show statistical significance.

**Questions:**

1. How well does ASL generalize beyond search-based QA? Could the same framework be applied to domains like coding?
2. Since the GRM is identified as a bottleneck, have you explored ways to scale or diversify it?
3. How do you ensure that the synthetically generated tasks remain diverse, realistic, and representative of real-world challenges?

---

> ### Author Response · Authors · 2025-11-18
>
> ## Thank you very much for reviewing our paper. Below are our responses to your comments.
>
> ### Regarding strengths
> + Thank you very much for recognizing the proactivity (reliance on very little human data) and scalability of our learning framework!
>
> ### Regarding weaknesses
> + W1: Thank you very much for your question! This is because our paper mainly conducts further research based on Search-r1 [1], and therefore focuses the study setting on the Deep Search scenario (as reflected in the paper’s title and in line 65 of the Introduction). Since search is a very important and well-defined task setting, many contemporaneous works ([1] [2] [3]) also conduct validation on such tasks. Thank you for your valuable suggestion; we will consider extending the ASL framework to more agentic task scenarios in future work.
> + W2: This is a very worthwhile topic to discuss! Compared to Search-r1, our ASL framework adds training phases for the Prompt Generator and the Generative Reward Model (GRM), which inevitably introduces extra computational overhead, but we believe it is well justified, as shown in Figure 4:
>     - During training of the Prompt Generator and GRM, since the three roles share parameters, the Deep Search capability of the Policy Model is also enhanced, reflected in the flatter portions of the test curve.
>     - Although ASL introduces additional computational overhead, it achieves performance comparable to the best performance of Search-r1 (at 2800 steps) at around 3800 steps. This means ASL requires less than 50% additional training steps to completely eliminate reliance on labeled data and RLVR and achieve self-learning, which we believe is a very worthwhile cost.
> + W3: Thank you very much for your suggestion. The accuracy shown in the figure has already been averaged over 8 test datasets (lines 334–338), so the improvement has considerable statistical significance and indicates the enhancement of our trained Agent’s capability on the Deep Search task. Following your suggestion, we will further supplement the paper with results run under multiple random seeds to further demonstrate this method’s significant advantages over other self-learning methods and its ability to reach Search-r1 performance without human data. As agentic RL training is time-consuming, we will submit the experimental results at a later date. Thank you again for your suggestion.

---

> ### Author Response · Authors · 2025-11-18
>
> ### Regarding questions:
> + Q1: You raised a very valuable question! Our proposed ASL method does not make any assumptions about tool types, nor does it incorporate any properties unique to the Search environment in the framework design. Therefore, we believe the ASL method can be extended to more agentic task scenarios. For example, one of the baselines compared in this paper, R-zero [4], is validated in the mathematical reasoning setting, and Absolute Zero [5] is validated in the coding setting. Thank you for your valuable suggestion; we will make extending ASL to more agentic task scenarios one of our key future work goals.
> + Q2: This is a very worthwhile topic to discuss! We discuss this issue in detail in lines 426–464 of Section 4.4: Not training the GRM leads to its discriminative preferences quickly being exploited by the Prompt Generator and the Policy Model, resulting in reward hacking. Therefore, we first experimentally confirmed that self-learning of the GRM within the ASL framework effectively expands its capability ceiling, helping it acquire increasingly better discriminative ability (the green bars in Figure 5 also demonstrate this). However, because the synthesized discriminative problems become increasingly difficult to break through the GRM’s existing capability boundary, we can, in practice, introduce a very small amount of real data (line 454) for late-stage GRM training, which can further raise the ceiling of the entire ASL system and allow it to continue iterative improvement. We believe this is an effective practical scheme to combine the human-in-the-loop paradigm with the ASL framework and further scale the entire system with minimal human effort.
> + Q3: This is a very interesting question! We are also curious whether the Prompt Generator can, like teachers in human society, progressively generate increasingly difficult, realistic, and diverse problems as students’ abilities improve. Based on your suggestion, we conducted the following case study to demonstrate the evolution of the Prompt Generator:
>
>     - Iteration 1:
>          +Q: Which Israeli city is known as the holiest city in Judaism?
>          +A: Jerusalem
>     - Iteration 2 (harder—depth):
>           +Q: Which most important religious site is located in the holiest city in Judaism?
>           +A: The Western Wall
>     - Iteration 3 (harder—breadth):
>           +Q: What is the name of the most expensive hotel in the city that contains the Western Wall, located in the religious capital of Israel, which is surrounded by the claimed capital of the State of Palestine?
>           +A: Park Hyatt Jerusalem
>
> As can be seen, ASL indeed produces increasingly difficult, realistic, and diverse questions during multi-role iterations.
> Thank you again for your valuable suggestion. We will include the above analyses and discussions in the revised version to better showcase the ASL framework’s enhancement of the Prompt Generator’s capabilities.
>
> ### Summary
> + Once again, thank you for your thoughtful and insightful suggestions; they greatly help improve the quality of our discussions on the paper’s motivation and results. We would be very happy to continue the conversation with you!
>
>
> [1] Jin B, Zeng H, Yue Z, et al. Search-r1: Training llms to reason and leverage search engines with reinforcement learning[J]. arXiv preprint arXiv:2503.09516, 2025.
>
> [2] Song H, Jiang J, Min Y, et al. R1-searcher: Incentivizing the search capability in llms via reinforcement learning[J]. arXiv preprint arXiv:2503.05592, 2025.
>
> [3] Lu H, Wen Y, Cheng P, et al. Search Self-play: Pushing the Frontier of Agent Capability without Supervision[J]. arXiv preprint arXiv:2510.18821, 2025.
>
> [4] Huang C, Yu W, Wang X, et al. R-Zero: Self-Evolving Reasoning LLM from Zero Data[J]. arXiv preprint arXiv:2508.05004, 2025.
>
> [5] Zhao A, Wu Y, Yue Y, et al. Absolute zero: Reinforced self-play reasoning with zero data[J]. arXiv preprint arXiv:2505.03335, 2025.

---

> ### Author Response · Authors · 2025-11-24
>
> Thank you once again for reviewing our paper and for the time and effort you devoted! We have uploaded the revised manuscript via the Rebuttal Revision. In response to your question Q3, we have presented and analyzed the evolution of the generative task in Appendix C.

---

> ### Author Response · Authors · 2025-11-27
>
> Dear Reviewer UEib, thank you again for your time and effort in reviewing our manuscript. We hope our responses have adequately addressed your concerns. We are happy to engage in further discussion if you have any additional questions.

---

### Meta-Review · Area_Chair_ynV4 · 2025-12-19

**Summary:**

All four reviewers raised concerns about statistical significance of the results, given that only a single random seed was used. While the authors promised to include more seeds in the future, the paper in its current state has just a single random seed. Reviewers also asked questions about the sensitivity to the initial policy (WoRw), robustness to reward hacking (q5p7), and for clarifications about the method/experimental details. The rebuttal provided answers to these questions, but did not provide data/evidence to support the claims.

**Reviewer Concerns:**

(see below)

**Reviewer Scores:**

UEib: 4 --> 4
*  [-] experiments are confined to a text-based search QA environment: authors mentioned that similar papers also evaluate on these tasks.
* [/] Complex training loop: authors note that their method improves training efficiency, and that parameter sharing somewhat decreases the method's complexity
* [-] Are results statistically significant? Authors mention they only used one random seed and didn't provide any statistical significance test. The plots in the paper are missing errors bars, making it hard to guess whether the differences are statistically significant.

q5p7: 4 --> 4
* [-] does the method avoid reward hacking, and how might this be measured? Authors argue that verifying the LM-proposed prompt with another LM (the generative reward model) should avoid this. They also argue that the stage-wise training procedure should help avoid certain failure modes (e.g., where it seems like policy is doing worse simply because it is answering more challenging questions). I found these answers reasonable, but they do not seem to provide concrete evidence/metrics that reward hacking is is isn't happening. Perhaps one thing the authors could do is evaluate on some tasks where we also has a verifiable reward, as a diagnostic to check for reward hacking.
* [-] missing information about random seeds: authors promised to do this, but the current paper has not been revised

WoRw: 6 --> 6(*)
* [+] Why does increasing entropy across the batch make sense? Authors clarified that it is entropy of answers for a single task, not averaged across a batch.
* [-] How is the prompt generator's difficulty calibrated? Authors promised to add the details to revisions in the future.
* [+] Some missing details about how rewards/scores are generated, whether parameters are shared, which environments were used, experimental details. Most of these were clarified in the rebuttal
* [/] Why was VeRL selected as the base algorithm? Authors said it was popular and easy to use.
* [-] How dependent is the performance on the initial policy? Authors speculated that it should be robust
* [-] Missing a preliminaries section. Authors promised to do this in a future revised version.
* [-] Sensitivity to hyperparameters: authors provided intuition for some, but didn't provide additional evidence/data.
* [-] Missing confidence intervals: not provided
* [/] Typesetting feedback (font sizes, citation format, etc): authors promised to revise in the future.

(*) Given that the initial score was 6, I do not think the reviewer would have further raised their score. However, based on the content of the review itself, this review seems more in line with a weak reject (4) rather than a weak accept (6), and would remain a "4" even after the rebuttal.


JUBh: 2 --> 2
* [-] missing details on random seeds and statistical significance: authors acknowledge that they have only used a single seed, and are working on getting more seeds.
* [/] adding a discussion of parameter scaling in other areas of RL: authors will address in the future
* [/] figure formatting: authors will address in the future
* [/] minor proofreading

---

### Decision · Program_Chairs · 2026-01-26

Reject